# Vision ELECTRA: Adversarial Masked Image Modeling with Hierarchical Discriminator

## Abstract

As a practical pre-training strategy for natural language processing (NLP), ELEC-TRA first masks parts of input texts and trains a generator and discriminator to reconstruct the texts and identify which parts are original or replaced. In this work, we propose Vision ELECTRA, namely $\mathcal{VE}$, which migrates ELECTRA to the vision domain with a non-trivial extension. Like ELECTRA, $\mathcal{VE}$ first leverages MAE or SimMIM to reconstruct images from masked image patches by generation. Particularly, random Gaussian noise is induced into the latent space of the generator to enhance the diversity of generated patches, in an adversarial autoencoding manner. Later, given original images and the reconstructed ones, $\mathcal{VE}$ trains an image encoder (usually ViT or Swin) via a hierarchical discrimination loss, where the discriminator is expected to (1) differentiate between original images and the reconstructed ones and (2) differentiate between original patches and generated ones. It gives $\mathcal{VE}$ a unique advantage that learns contextual representations characterizing images in both macro- and micro-levels (i.e., the entire image and individual patches). Extensive experiments have been carried out to evaluate $\mathcal{VE}$ with baselines under fair comparisons. The findings demonstrate that $\mathcal{VE}$ based on the ViT-B attains a top-1 acc of 83.43% on the ImageNet-1K image classification task with a 1.17% improvement over baselines under continual pre-training. When transferring $\mathcal{VE}$ pre-trained models to other CV tasks, including segmentation and detection, our method surpasses other methods, demonstrating its applicability on various tasks.

## 1 Introduction

Self-supervised pre-training strategies surge nowadays, resulting in powerful pre-trained models, such as BERT (Kenton & Toutanova, 2019), GPT (Radford et al., 2018), and MAE (He et al., 2022), for various tasks. Among these strategies, masked autoencoding strategies have been widely adopted by numerous solutions, including the masked language model (MLM) (Salazar et al., 2020) for natural language processing (NLP) (Strubell et al., 2019) and the masked image model (MIM) (Xie et al., 2022) for computer vision (CV). In general, these strategies first mask part of input images/texts, then learn to generate the masked ones in the context of masking and reconstruct the images/texts. To further improve masked autoencoding for NLP, ELECTRA (Clark et al., 2020) has been proposed to follow up the MLM with a discriminator, where the MLM and discriminator are jointly trained to reconstruct the texts and identify which parts of texts are original or replaced. In contrast to the vanilla MLM, ELECTRA outputs the text encoder of discriminator as the outcome of self-supervised pre-training. Earlier studies (Clark et al., 2020) show that such text encoder of discriminator could outperform BERT in learning contextual representation of texts.

Encouraged by the success of ELECTRA, efforts have been done to enhance masked image models for CV. For example, He et al. (2022) proposed Masked Autoencoder (MAE) that trains vision transformers to reconstruct images using part of image patches, where the encoder of network is adopted as a scalable vision learner by self-supervision. Further, to lower the training cost of MIM, Xie et al. (2022) introduces SimMIM that incorporates random masking on image patches and raw pixel regression loss with light-weight prediction heads. More recently, Fei et al. (2023) studies to incorporate MAE within the training framework of generative adversarial networks (GANs), where a discriminator is introduced to replace the loss of pixel-wise regression for the image reconstruction task. Though these works have gathered the necessary ingredients, such as masking strategies,

autoencoders, reconstruction losses, and even discriminators to identify original/reconstructed images, they all fail to capture the key concept of ELECTRA for computer vision tasks-leveraging the encoder of discriminator rather than that of autoencoders as the outcome of pre-training. The non-trivial design of ELECTRA's discriminative task for computer vision is that the model learns from all input tokens, rather than just a small masked-out subset, granting an appreciable depth to image comprehension (Clark et al., 2020). It is thus reasonable to replicate ELECTRA for potential performance enhancement in self-supervised pre-training of images.

In this work, we aspire to extend the ELECTRA model to the field of computer vision through substantial enhancements, culminating in the proposal of Vision ELECTRA, colloquially referred to as $\mathcal{VE}$. Mirroring the operational framework of ELECTRA, $\mathcal{VE}$ initiates its process by employing either MAE or SimMIM to regenerate images from masked image patches via generation. Specifically, random Gaussian noise is injected into the latent space of the generator to diversify the assortment of created patches while adhering to the principles of adversarial autoencoding. Subsequently, $\mathcal{VE}$ implements an image encoder, typically ViT or Swin, as the image discriminator employing a hierarchical discrimination loss. Within the joint training procedure of generator and discriminator for $\mathcal{VE}$, the discriminator juggles two key responsibilities. Firstly, it distinguishes between the original images and their reconstructed counterparts. Secondly, it discerns between the original patches and those that have been generated. The use of hierarchical discrimination loss accords $\mathcal{VE}$ a distinct advantage by imparting the ability to learn contextual representations that characterize images at both macro- and micro-levels. In other words, it can understand both the overall image and its individual patches, granting an appreciable depth to image comprehension.

The main contributions are as follows: (1) We propose a novel MIM framework $\mathcal{VE}$, following core concepts of ELECTRA that adopt the *generator-discriminator* paradigm for CV and leverage the encoder of discriminator as the pre-training outcome. (2) The proposed $\mathcal{VE}$ incorporates three innovative designs: i) adversarial pre-training of the generator to enhance "image authenticity", ii) incorporation of Gaussian noise to perturb the latent space and thus diversify the reconstructed images, iii) introduction of the hierarchical discriminator to capture contextual representations of images at both macro- and micro-levels. (3) We perform extensive experiments demonstrating remarkable performance superiority compared to mainstream MIM methods in downstream tasks.

## 2 RELATED WORK

**Mask Image Modeling**  Self-supervised learning, widely used in NLP tasks (Brown et al., 2020; Kenton & Toutanova, 2019), has found success with the adoption of pixel sequences for prediction (iGPT) (Chen et al., 2020) and masked token prediction for self-supervised pre-training (ViT) (Dosovitskiy et al., 2020). Following these advancements, Transformer-based architectures have emerged in Masked Image Modeling (MIM) (Bao et al., 2021; Feichtenhofer et al., 2022; He et al., 2022; Xie et al., 2022; Wang et al., 2022; Wei et al., 2022). MIM models predict masked content within visible regions, enriching visual interpretations. BEiT (Bao et al., 2021) has enhanced this area by learning via discrete token prediction. MAE (He et al., 2022) and SimMIM (Xie et al., 2022), meanwhile, favor pixel-wise masking and reconstruction, eliminating the need for discrete token representations.

**Generative Adversarial Networks**  (Goodfellow et al., 2014) have proven effective in generating high-quality artificial data. The practice of using the discriminator of GAN in subsequent operations is akin to our approach and was introduced by Radford et al. (2015). Similarly, MaskGAN (Fedus et al., 2018) trains its generator to fill in removed tokens, a concept paralleling MIM. More recently, Fei et al. (2023) proposed to incorporate GANs in the MAE framework to replace the loss of pixel-wise regression for the enhanced image reconstruction task.

**ELECTRA**  (Clark et al., 2020) is a two-part model with a generator and a discriminator, both based on BERT (Kenton & Toutanova, 2019). The generator uses MLM to find replacements for a MASK token and the discriminator detects these replacements in the text. After pre-training, the generator is discarded, making the discriminator a pre-trained language model. ELECTRA either outperforms BERT with the same computing power or performs similarly.

**Discussion**  Our work presents several unique contributions compared to previous studies. In terms of novelty, this work is the first to migrate ELECTRA (Clark et al., 2020) into CV. Non-trivial extensions, such as adversarial pre-training of generator with Gaussian noises in the latent space, alongside a hierarchical discrimination loss for representation learning at both macro- and micro-levels,

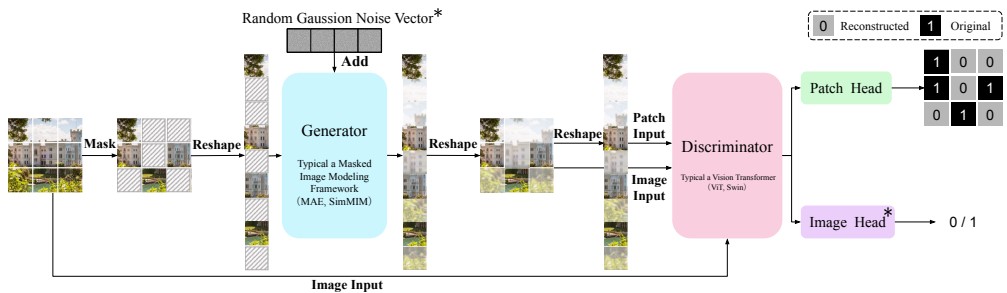

Figure 1: The overall design of $\mathcal{VE}$. Note that the discriminator takes both the reconstructed patches/image and the original patches/image as inputs, while they have different labels. '*' means the non-trivial components, which are never used before.

allow $\mathcal{VE}$ to outstrip a straight re-implementation of ELECTRA for CV. Unlike Fei et al. (2023), who also trains an MAE generator model with a discriminator via the GAN-like loss, $\mathcal{VE}$ employs the encoder of the discriminator rather than the MAE as the outcome of pre-training, following the core design of ELECTRA, where the discriminative task can allow the model learns from all input tokens, rather than just a small masked-out subset, making it grant an appreciable depth to image comprehension. Further, $\mathcal{VE}$ improves upon simple GAN Loss with hierarchical discrimination, enhancing the generator/discriminator's ability to reconstruct and identify patches/images. All these innovative designs make $\mathcal{VE}$ a novel and effective framework for pre-training on top MIM.

## 3 METHODOLOGY

In this section, we first present the overall design of $\mathcal{VE}$, then introduce the key algorithm designs for training the generator and discriminator within $\mathcal{VE}$.

### 3.1 OVERALL DESIGN OF VISION ELECTRA

As shown in Figure 1, $\mathcal{VE}$ is built upon a *generator-discriminator* paradigm, where the image encoder of the discriminator is the outcome of pre-training. In our study, the generator $\mathcal{G}$ is a masked image modeling (MIM) model (e.g. MAE (He et al., 2022), SimMIM (Xie et al., 2022)), which masks a portion of input images and predicts the masked patches. In the meanwhile, the discriminator $\mathcal{D}$ is a vision transformer (e.g. ViT (Dosovitskiy et al., 2020), Swin-Transformer (Liu et al., 2021)), which classifies each image patch or the entire image is original or reconstructed.

For the $\mathcal{G}$, it takes the visible image patches as input, obtaining the latent codes and then reconstruct the masked patches from the latent codes by a small network. Different from MAE, $\mathcal{G}$ introduces a Gaussian noise vector to perturb the latent space, mildly inhibiting the capabilities while enhancing feature diversity (Tian et al., 2020), thus strengthening the $\mathcal{D}$ in an adversarial autoencoding manner.

For the $\mathcal{D}$, it serves a dual role: distinguishing between original and reconstructed patches / images. These tasks share the backbone weights, with different task-specific heads. In alignment with established strategies (He et al., 2022; Xie et al., 2022), $\mathcal{D}$ similarly process the patch tokens through a sequence of Transformer blocks. Patch discrimination is facilitated by a CNN-based head, discerning tokens in the data from those replaced by $\mathcal{G}$. For image discrimination, a linear projection layer is utilized as the image head, determining whether the image is original or reconstructed.

Given an image $I$, we first split it into non-overlapping patches and add a CLS token $x_0$, i.e., $I = \{x_0, x_1, \cdots, x_N\}$, then we randomly mask some patches with a probability, for example 0.75 in MAE. After that, we use $\mathcal{G}$ to reconstruct the masked patches, and finally, we feed the reconstructed and the original images to $\mathcal{D}$ for discrimination (see Figure 1).

### 3.2 GENERATOR VIA ADVERSARIAL MASKED IMAGE MODELING

$\mathcal{G}$ reconstructs images from masked image patches using a MIM framework.

**Patch Generation by Pixel-wise Regression** The output of $\mathcal{G}$ is reshaped to form the reconstructed image. Similar to the previous MIM framework (He et al., 2022; Xie et al., 2022), the loss function used in our approach computes the MSE between the reconstructed and original images in the pixel space. To calculate the loss, we only consider the masked patches, following a similar

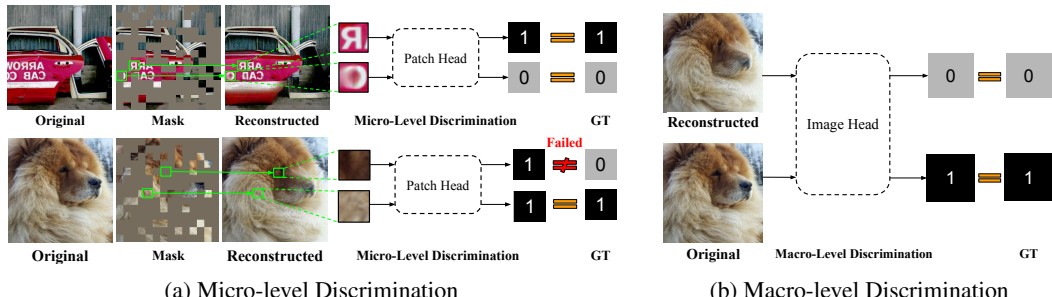

(a) Micro-level Discrimination          (b) Macro-level Discrimination

Figure 2: An Illustration of Hierarchical Discrimination. At the micro-level, the adversarial training will progressively make the reconstructed patches indistinguishable from patch head. At the macro-level, the image head introduces another discriminative constraint for the pipeline.

approach as MAE. The definition of our loss function (patch_loss) is as follows.

$$\mathcal{L}_{\text{Img}}(\theta_G) = MSE(x^r, x^m) = \frac{1}{M} \sum_{i=1}^{M} (x_i^m - x_i^r)^2 \ , \tag{1}$$

where $x^r$ represents the patches recovered by $\mathcal{G}$'s decoder using the latent code $z$, i.e., $x^r = f(z+\tilde{z})$, $\tilde{z}$ represents the random Gaussian noise vector, $x^m$ represents the masked patches of original image $x$ and $M$ represents the number of patches.

**Image Reconstruction in Adversarial Manner**    It is easy to discriminate the reconstructed image from the original one for the discriminator, if the reconstructed patches solely rely on the patch loss, since the reconstructed patches are normally blurry. Therefore, the simple task cannot benefit the discriminator to capture useful information for downstream tasks. To address this issue, we introduce an adversarial loss (Goodfellow et al., 2014) to enhance the authenticity of the generated images. The reconstructed images fool the discriminator in two levels – macro level and micro level, i.e, the discriminator should treat the entire reconstructed image and each reconstructed patch as real ones, resulting in an adversarial loss as follows:

$$\mathcal{L}_{\text{GAN}}(\theta_G) = -\frac{1}{M} \sum_{i=1}^{M} \log\left(\mathcal{D}(x_i^r)\right) - \log\left(\mathcal{D}(x_0^r)\right) \tag{2}$$

where $x_i^r$ denotes the $i$th reconstructed patch and $x_0^r$ denotes the CLS token, the representation of which denotes the entire reconstructed image, and the input entire image is composed of the reconstructed and original unmasked patches. $\mathcal{D}(x_i^r)$ denotes the predicted label of the $i$th patch by the discriminator $\mathcal{D}$, likewise, $\mathcal{D}(x_0^r)$ is the prediction of the entire reconstructed image. $M$ denotes the number of reconstructed patches in an image.

### 3.3 Discriminator via Hierarchical Discrimination Loss

$\mathcal{D}$ is responsible for distinguishing between original and reconstructed inputs. It achieves this task by employing a weight-shared backbone and two sub-task heads.

**Micro-Level Discrimination**    We consider the patch head as the micro-level discrimination, which classifies whether a patch is original or reconstructed. Therefore, we use a binary cross-entropy loss, i.e.,

$$\mathcal{L}_{\text{Patch}}(\theta_D) = -\frac{1}{N} \sum_{i=1}^{N} [y_i \log(\mathcal{D}(x_i)) + (1 - y_i) \log(1 - \mathcal{D}(x_i))] \tag{3}$$

where $x_i$ is an image patch and $y_i$ is the corresponding label. For the original patch in an image, $y_i = 1$, and $y_i = 0$ for a reconstructed patch. $N$ represents the number of patches in image $I$. Note that $I$ can be the reconstructed or original image.

**Macro-Level Discrimination**    The macro-level discrimination is to classify whether the entire image is original or not. Similar to micro-level discrimination, we also use the binary cross-entropy loss. The difference is that we use the representation of CLS token to compute the loss function, i.e.,

$$\mathcal{L}_{\text{CLS}}(\theta_D) = -y \log(\mathcal{D}(x_0)) - (1 - y) \log(1 - \mathcal{D}(x_0)) \tag{4}$$

where $y = 0$ for a reconstructed image and $y = 1$ for the original image.

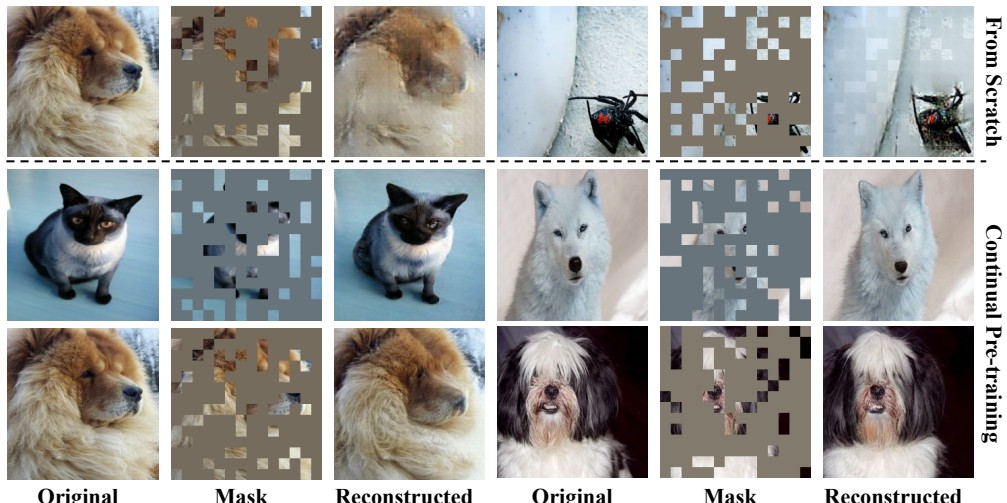

Figure 3: Illustration of Reconstructed Images with Two Pre-training Schemes.

| Method | Pre-training strategies | Input size | Mask ratio | Top-1 acc (%) |
|---|---|---|---|---|
| SimMIM | 100ep from Scratch | $224^2$ | 60% | 81.39 |
| MAE | 100ep from Scratch | $224^2$ | 75% | 81.68 |
| $\mathcal{VE}$ | 100ep from Scratch | $224^2$ | 75% | 80.73 |
| SimMIM | Official | $224^2$ | 60% | 82.16 |
| MAE | Official | $224^2$ | 60% | 82.89 |
| SimMIM | Official + 50ep | $224^2$ | 60% | 82.26 |
| MAE | Official +50ep | $224^2$ | 75% | 83.06 |
| $\mathcal{VE}$ | Official +50ep | $224^2$ | 75% | **83.43** |

Table 1: Performance Comparisons: '100ep' means pre-training from scratch for 100 epochs, 'Official' refers to fine-tuning based on the official release of pre-trained models, '+50ep' refers to an additional 50 epochs pre-training based on the official releases (He et al., 2022; Xie et al., 2022).

### 3.4 JOINT PRE-TRAINING OF GENERATOR AND DISCRIMINATOR

We use the following loss to jointly train the generator and discriminator,

$$\mathcal{L} = \min_{\theta_G, \theta_D} \mathcal{L}_{Img}(\theta_G) + \lambda \mathcal{L}_{GAN}(\theta_G) + \mathcal{L}_{Patch}(\theta_D) + \mathcal{L}_{CLS}(\theta_D) \tag{5}$$

To stabilize the training process, we use a small $\lambda$ in our experiments, i.e, $\lambda = 0.2$.

Note that all tokens are visible to the discriminator, which is the same as downstream tasks, narrowing the gap between pre-training and fine-tuning.

## 4 EXPERIMENT

We employ the MAE (He et al., 2022) as the $\mathcal{G}$ within the $\mathcal{VE}$. For the $\mathcal{D}$ component of $\mathcal{VE}$, we utilize the ViT-B/16 (Dosovitskiy et al., 2020). Our experimental configuration entails self-supervised pre-training using the ImageNet-1K (Deng et al., 2009). Subsequently, we also have fine-tuned downstream tasks such as classification (Lu & Weng, 2007), segmentation (Guo et al., 2018), and detection (Zou et al., 2023). More experimental settings are provided in the supplementary material.

### 4.1 COMPARISONS WITH PREVIOUS RESULTS ON IMAGE CLASSIFICATION

For fair comparison, following previous works (He et al., 2022; Xie et al., 2022), we conduct experiments using ViT-B. The quantitative results are shown in Table 1 and the qualitative results are exhibited in Figure 3. Specifically, two sets of comparisons are as follows.

**Pre-training from Scratch** We train $\mathcal{VE}$ using 100 epochs from its random scratch. For SimMIM (Xie et al., 2022) and MAE (He et al., 2022), we also perform the same 100 epochs pre-training

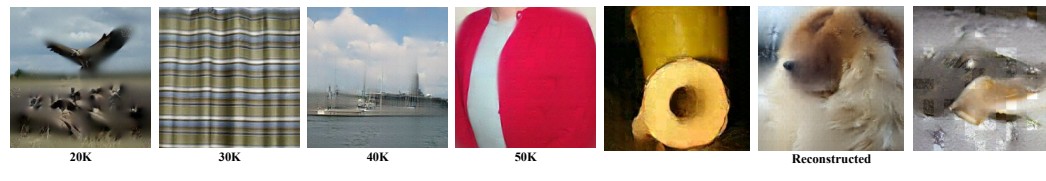

| 20K | 30K | 40K | 50K | | Reconstructed | |

(a) Sharing Weight          (b) Smaller Generator

Figure 4: Examples of Reconstructed Images. (a) Sharing Weight: reconstructed images at varying training steps, (b) Smaller Generator: reconstructed images with a smaller generator.

| Settings | Pre-training strategies | Input size | Top-1 acc (%) |
|---|---|---|---|
| $\mathcal{VE}$ (Sharing Weight) | Official+50ep | $224^2$ | 82.37 |
| $\mathcal{VE}$ (Smaller Generator) | Official+50ep | $224^2$ | 81.93 |
| $\mathcal{VE}$ | Official+50ep | $224^2$ | **83.43** |

Table 2: Experiment Results on Model Exploration.

from scratch with the official configuration. We then finetune these models using additional 100 epochs and compare their performance. As shown in Table 1, our method obtains competitive results compared to the mainstream MIM methods ($\mathcal{VE}$: 80.73 vs. SimMIM: 81.39 vs. MAE: 81.68).

We believe the $\mathcal{VE}$ performs marginally worse than baselines due to the following two reasons. (1) $\mathcal{VE}$ leverages an adversarial loss derived from GAN to ensure the quality of image reconstruction while the GAN training sometimes is difficult to converge and frequently leads to model collapse (Salimans et al., 2016). (2) The discriminator in $\mathcal{VE}$ could not learn good representations from distinguishing between original images and low-quality generated ones (please refer to the examples in Fig 3), while high-quality image generation usually needs more efforts to train (especially compared to the cost of language generation tasks in vanilla ELECTRA). Note that we use 100 training epochs here to follow the settings of vanilla ELECTRA (Clark et al., 2020).

**Continual Pre-training**    In addition to training from scratch, we adopt pre-trained models from their official releases and conduct experiments through *continual-pre-training* these models. Specifically, here, we first build-up a $\mathcal{VE}$ model using ViT-B (as the discriminator $\mathcal{D}$) pre-trained by SimMIM, and continue to train $\mathcal{VE}$ using additional 50 epochs. We then compare the discriminator of such $\mathcal{VE}$ model with SimMIM and MAE under the same continual pre-training settings, where both SimMIM and MAE were firstly loaded from official release and further trained with additional 50 epochs. As shown in Table 1, while both SimMIM and MAE could be improved by continual pre-training, our proposed method still outperforms these models and achieves a Top-1 acc of 83.43%. Compared to MAE (Official+50ep), $\mathcal{VE}$ exhibits an improvement of 0.37 points. Compared to the SimMIM (Official+50ep), $\mathcal{VE}$ demonstrates a superior performance, surpassing it by 1.17 points. Note that, in this setting, the discriminator of $\mathcal{VE}$ was derived from ViT-B pre-trained by SiMIM. Our method demonstrates absolute performance improvements with 50 additional epochs of continual-pre-training, surpassing both SimMIM (Official) and SimMIM (Official+50ep). Furthermore, as illustrated in Figure 3, it is evident that the presence of authentic reconstructed images obviously enhances the performance of the hierarchical discrimination across both macro- and micro-levels.

## 4.2 EMPIRICAL STUDIES ON MODEL EXPLORATION

Inspired by the *Model Extension* section of ELECTRA (Section 3.2 of Clark et al. (2020)), we discuss several options of $\mathcal{VE}$. Particularly, we focus on two settings: (a) Sharing Weight and (b) Smaller Generator, here.

**Sharing Weight**    As was mentioned, both generator $\mathcal{G}$ and discriminator $\mathcal{D}$ in $\mathcal{VE}$ are derived from the same model, i.e., ViT-B as their encoders. It is reasonable to assume sharing weights between these two encoders might be able to improve the performance under the same training budget. As shown in Table 2, the performance of $\mathcal{VE}$ under the sharing weight setting lags behind the vanilla $\mathcal{VE}$ (82.37 vs. 83.43). We consider the degradation is due to the disparity between task domains of generator and discriminator–sharing weights between these two would lead to instability during

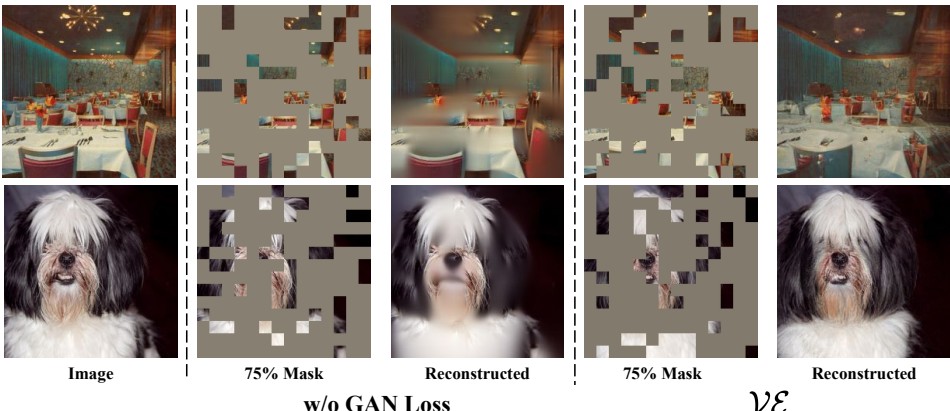

Figure 5: Examples of Recovered Images with and without GAN Loss within $\mathcal{VE}$.

pre-training. Figure 4(a) shows that with increasing training steps, the generator under sharing weight setting produces images of varying quality. Some are with high-quality, while others are notably inferior. This variability underscores our previous assumption that weight sharing leads to instability during pre-training. Hereby, we refrain from the sharing weight strategy in $\mathcal{VE}$.

**Smaller Generator** As the pre-training outcome of $\mathcal{VE}$ is the encoder of discriminator, it is reasonable to doubt the use of smaller generator (with fewer parameters) could still achieve good performance. To be specific, compared to ViT-B used by the vanilla $\mathcal{VE}$, we employ ViT-S as the encoder for the generator to build up $\mathcal{VE}$ (Smaller Generator). Specifically, we first pre-train ViT-S within MAE using 200 epochs and use such ViT-S model as the encoder of generator in $\mathcal{VE}$ (Smaller Generator). We still adopt ViT-B pre-trained by SimMIM as the encoder of discriminator in $\mathcal{VE}$ (Smaller Generator) and perform an additional 50 epochs of pre-training. In Table 2, it is evident that $\mathcal{VE}$ (Smaller Generator) performs worse than vanilla $\mathcal{VE}$ in terms of Top-1 acc (81.93 vs. 83.43). The performance degradation is due to the low-quality of images generated by the smaller generator (shown in Figure 4(b)), when employing ViT-S as the encoder of generator.

### 4.3 ABLATION STUDY

The ablation study is conducted to validate the effectiveness of individual components within our $\mathcal{VE}$. All experiments in this section are pre-trained on the ImageNet-1K dataset (Deng et al., 2009) and subsequently fine-tuned on the Image-1K image classification task.

**Patch Loss** As presented in Table 3, it is evident that by excluding the patch loss, the Top-1 acc of $\mathcal{VE}$, fine-tuned for the ImageNet-1K image classification task, undergoes a marginal reduction from 83.43 to 83.26. This degradation is probably due to the ignorance to the patch-level discrimination, as the discriminator without patch loss solely appraises the overall image authenticity (original or reconstructed). As a result, $\mathcal{VE}$ w/o the patch loss would fail to train the encoder of discriminator with ability of representation at micro-level.

**GAN Loss** As shown in Table 3, it shows that by excluding the GAN loss, the Top-1 acc of $\mathcal{VE}$, fine-tuned for the ImageNet-1K image classification task, experiences a slight decrement, shifting from 83.43 to 83.24. Additionally, the qualitative results, presented in Figure 5, demonstrate the images reconstructed by $\mathcal{VE}$ and $\mathcal{VE}$ w/o GAN Loss. It is obvious that the GAN Loss can help $\mathcal{VE}$ to generate images with higher authenticity. As a result, $\mathcal{VE}$ with the GAN Loss could train the encoder of discriminator with better capacity of feature learning at macro-level.

Yet, ablation studies on the patch and GAN losses have already proved the effectiveness of our proposed hierarchical discrimination in both image generation and contextual representation at both micro/macro-levels. The joint training of generator and discriminator in adversarial settings could be benefited from both losses.

**CLS Loss** As depicted in Table 3, it is evident that the inclusion of the CLS Loss yields a substantial performance enhancement, notably increasing the Top-1 acc from 83.01 to 83.43. This

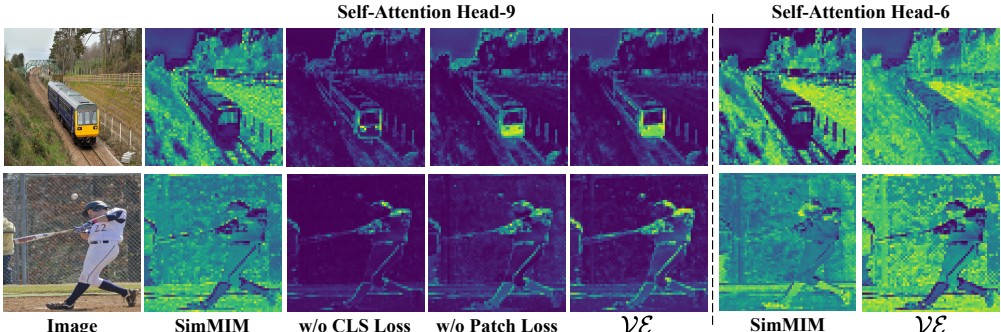

Figure 6: Visualization of Self-attention from Pre-trained Models. We have visualized self-attention maps on the two heads separately. The figures of self-attention head-9 demonstrate micro-level representation ability, while self-attention head-6 exhibits macro-level representation ability.

| Methods | Input size | Pre-training strategies | Patch loss | GAN loss | CLS loss | Gaussian noise | Top-1 acc (%) |
|---|---|---|---|---|---|---|---|
| SimMIM | $224^2$ | Official+50ep | | | | | 82.26 |
| $\mathcal{VE}$ | $224^2$ | Official+50ep | | ✓ | ✓ | ✓ | 83.26 |
| $\mathcal{VE}$ | $224^2$ | Official+50ep | ✓ | | ✓ | ✓ | 83.24 |
| $\mathcal{VE}$ | $224^2$ | Official+50ep | ✓ | ✓ | | ✓ | 83.01 |
| $\mathcal{VE}$ | $224^2$ | Official+50ep | ✓ | ✓ | ✓ | | 83.33 |
| $\mathcal{VE}$ | $224^2$ | Official+50ep | ✓ | ✓ | ✓ | ✓ | **83.43** |

Table 3: Ablation Study. For fair comparison, we use the ViT-B pre-trained using above methods under the same setting and finetuned on the downstream task for experiments.

improvement can be attributed to the integration of the CLS token, which serves as an image head for pre-training the discrimination between original and reconstructed images. This feature proves highly beneficial for the subsequent fine-tuning in image classification. The positive impact the CLS token is also illustrated in the qualitative results presented in Figure 6. Additionally, drawing inspiration from Dino (Caron et al., 2021), we also visualize the attention heat maps for the final layer of ViT-B. Figure 6 clearly demonstrates that, following the incorporation of CLS Loss and CLS token, $\mathcal{VE}$ exhibits a strengthen focus on the primary subject within the image.

**Gaussian Noise** In Figure 7, we highlight the differences among several kinds of images: the original image, the diverse images generated by $\mathcal{VE}$ with and without Gaussian noise. We can see that using Gaussian noise is able to generate an image that is more different but still real. To some extent, adding Gaussian noise to the latent code improves the diversity of the generated images, which plays the role of data augmentation for the discriminator, hence, it can benefit pre-training. Looking at the quantitative results in Table 3, the Top-1 accuracy increases from 83.33 to 83.43 by using additional Gaussian noise.

**Discriminative Pre-training Strategy** Figure 6 reveals that, in contrast to SimMIM, which pays much attention to the entire image instead of objects, i.e., self-attention heads (head-6 and head-9) exhibit similar macro-level attention, the $\mathcal{VE}$ is able to demonstrate both macro- and micro-level attention, e.g, head-9 focuses on objects, while head-6 focuses on the whole image. This qualitative observation demonstrates that, unlike prevalent MIM methods like MAE and SimMIM, the pre-trained discriminator of $\mathcal{VE}$ obtains smaller task domain gap between the pre-trained tasks and the downstream tasks Xie et al. (2023), facilitating more effective fine-tuning and improving the performance (quantitative results as shown in Table 1).

## 4.4 CROSS-TASK TRANSFER LEARNING EXPERIMENTS

**Semantic segmentation** To maintain a fair comparison, we fine-tune the pre-trained models provided by MAE and SimMIM under the same configuration. Table 4(a) shows that our framework significantly improves the performance of semantic segmentation compared to the mainstream methods, e.g., by 2.06 points for SimMIM and 1.02 points for MAE. The reasons are two-fold. First, the

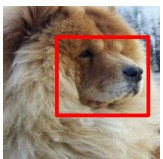 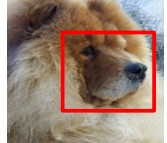 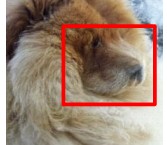 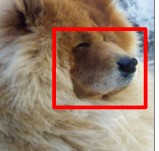 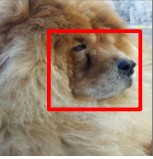 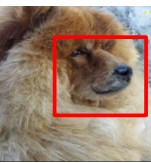

| **Image** | **w/o Guassian Noise** | $\mathcal{VE}$ **- Sample 1** | $\mathcal{VE}$ **- Sample 2** | $\mathcal{VE}$ **- Sample 3** | $\mathcal{VE}$ **- Sample 4** |

Figure 7: Examples of Recovered Images with and w/o Gaussian Noise within the $\mathcal{VE}$.

| Method | Pre-training strategies | Input size | mIoU |
|--------|------------------------|------------|------|
| SimMIM | Official+50ep | $512^2$ | 46.76 |
| MAE | Official+50ep | $512^2$ | 47.80 |
| $\mathcal{VE}$ | Official+50ep | $512^2$ | **48.82** |

(a) ADE20K Semantic Segmentation

| Method | Pre-training strategies | Input size | AP$^{box}$ |
|--------|------------------------|------------|------------|
| SimMIM | Official+50ep | $768^2$ | 45.95 |
| MAE | Official+50ep | $768^2$ | 46.10 |
| $\mathcal{VE}$ | Official+50ep | $768^2$ | **46.60** |

(b) COCO Object Detection

Table 4: Cross-Task Transfer Learning Experiments: (a) ADE20K semantic segmentation using UperNet. The reproduction code is from mae-segmentation (Li, 2022). (b) COCO object detection using a ViT Mask R-CNN baseline. The reproduction code is from MIMDet (Fang et al., 2022).

discriminator treats all patches as visible ones in the pre-training phase, which is the same as the fine-tuning phase. Second, during pre-training we use a patch-level classification task, to some extent, which is similar to pixel-level classification, benefiting the task of semantic segmentation.

**Object detection** Also, we fine-tune the pre-trained models provided by MAE and SimMIM under the same configurations for the task of object detection. As shown in Table 4(b), compared to mainstream methods, our $\mathcal{VE}$ performs better under the same configuration. The metric score obtained by our $\mathcal{VE}$ is 0.5 points higher than MAE (46.60 vs. 46.10, AP$^{box}$). Additionally, our $\mathcal{VE}$ also outperforms the another counterparts – SimMIM by 0.65 points (46.60 vs. 45.95). The reason for the improvement is that our pre-trained model achieves a better ability to localize objects, i.e., $\mathcal{VE}$ can pay attention to both the primary object and the entire scene, while the ViT pre-trained by SimMIM shows global attention (see Fig. 6).

### 4.5 REMARKS ON EXPERIMENT RESULTS

Here, we summarize above experiment results and make three conclusions as follows. (1) $\mathcal{VE}$ is effective in representation learning, and it can outperform SimMIM and MAE in continual pre-training settings with the same amount of training epochs (i.e., Official+50ep introduced in Section 4.1). (2) The joint training procedure employed $\mathcal{VE}$ trains both generator and discriminator simultaneously and adversarially – The higher the quality of images reconstructed by the generator, the more effectively the encoders are trained in the discriminator. (3) $\mathcal{VE}$ works well on various CV tasks, including classification, segmentation and detection, while every component proposed in $\mathcal{VE}$ has a specific role contributing to the overall functionality.

## 5 CONCLUSION

In this work, we propose a novel MIM framework $\mathcal{VE}$, following the core concepts of ELECTRA that adopt the *generator-discriminator* paradigm for CV and leverage the encoder of discriminator as the pre-training outcome. To achieve the goal, several non-trivial technical contributions have been made, including adversarial pre-training of the generator to enhance image authenticity, infusion of Gaussian noise in the latent space for image diversity, and a hierarchical discrimination loss that enables representation at both macro/micro-levels. Extensive experiments have been carried out the demonstrate the performance advancements of $\mathcal{VE}$ for downstream tasks. Our method, VE, exhibits superiority in performance over mainstream Mask Image Modeling (MIM) methods, including SimMIM and MAE, under fair comparisons. Empirical studies and ablation studies have explored the dimensions of our framework design and prove the soundness of every component proposed in our framework. Cross-task transfer learning experiments further confirm the applicability of $\mathcal{VE}$ on various CV applications including segmentation and detection. Being a first-of-its-kind initiative to leverage ELECTRA in CV, this work pioneers a unique blend of NLP techniques and CV models. Future works aim to expand on practical applications of this method and enhance the efficiency of pre-training further.

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
