# OpenReview forum: "Vision ELECTRA: Adversarial Masked Image Modeling with Hierarchical Discriminator"
_ICLR.cc/2024/Conference — Submitted to ICLR 2024_

### Official Review · Reviewer_XJEn · 2023-11-01

**Soundness:** 2 fair
**Presentation:** 2 fair
**Contribution:** 2 fair
**Rating:** 5
**Confidence:** 3

**Summary:**

This paper introduces VE, a MIM framework for computer vision (CV), inspired by ELECTRA. VE uses a generator-discriminator setup for pre-training, with the discriminator's encoder as the outcome. It adds innovations like adversarial training, Gaussian noise for diversity, and hierarchical discrimination for capturing macro and micro-level features. Experiments show VE outperforms mainstream MIM methods like SimMIM and MAE, especially in continual pre-training. VE excels in various CV tasks like classification, segmentation, and object detection. The paper validates each VE component empirically and highlights its cross-task transfer learning capabilities.

**Strengths:**

1. The paper introduces a Masked Image Modeling (MIM) framework, VE, which adapts ELECTRA's principles to the field of computer vision.

2. The framework incorporates some elements, including adversarial training for image quality, Gaussian noise for diversity, and hierarchical discrimination for capturing features at both macro and micro levels.

3. VE leverages a generator-discriminator setup and the encoder of the discriminator as the pre-trained model, demonstrating its effectiveness in representation learning.

4. The paper conducts empirical validations, confirming the effectiveness of each component within the VE framework.

5. VE exhibits cross-task transfer learning capabilities, showing its applicability in different CV applications.

**Weaknesses:**

1. As indicated in Table 1, VE lags noticeably behind SimMIM and MAE when trained from scratch. It's only in the Continual Pre-training setting that VE outperforms SimMIM and MAE. However, it's worth noting that Continual Pre-training still relies on SimMIM's model, which means that before using VE, SimMIM pre-training is still necessary, limiting the practicality of VE.

2. On page 6, the paper mentions, 'Note that, in this setting, the discriminator of VE was derived from ViT-B pre-trained by SiMIM.' However, the paper doesn't provide Continual Pre-training results based on pre-trained MAE models. This leaves us uncertain about VE's generalizability when based on MAE.

3. Comparing VE to SimMIM and MAE in a Continual Pre-training setup might seem somewhat unfair. This is because during the continued training phase, SimMIM and MAE are still engaged in generative pre-training, while VE follows a discriminative training approach. Therefore, it's not surprising that VE outperforms SimMIM and MAE in this context. An alternative and more equitable baseline would involve subjecting SimMIM and MAE to discriminative pre-training (e.g., contrastive learning training) during the continued training phase, and then comparing their performance to VE.

4. One important experiment that is missing is the result of using VE's MIM model as the derived pre-trained model.

5. The approach of jointly training the generator and discriminator as depicted in Equation 5 typically does not work for GANs. GANs usually employ an alternating training strategy, where the discriminator is trained first, followed by training the generator. Typically, the discriminator undergoes more updates than the generator to ensure stable training. I remain skeptical about the effectiveness of the joint training strategy for GANs outlined in Equation 5.

**Questions:**

My main concern is that VE outperforms SimMIM and MAE only in the Continual Pre-training setting, implying that VE still relies on other pre-training methods (the pre-training model used in the article is based on SimMIM). VE is not an independent pre-training solution like SimMIM and MAE.

---

### Official Review · Reviewer_rgaC · 2023-11-01

**Soundness:** 2 fair
**Presentation:** 3 good
**Contribution:** 2 fair
**Rating:** 5
**Confidence:** 4

**Summary:**

This paper adapts a pretraining framework widely used in NLP, ELECTRA, to the vision domain and shows empirically that the Vision ELECTRA performs better than the existing popular MIM methods.

**Strengths:**

The work adapts a significant pretraining framework to the vision domain, which would likely promote the advancement of research in the multimodal domain.

**Weaknesses:**

Given the simplicity of the proposed method, I expect much more extensive empirical studies to be conducted, so that we can learn how effective the ELECTRA framework is in the vision domain. For example, In Table 1, VE is only pretrained for a short period from scratch in addition to continual pretraining, neither of which is a common setting for self-supervised pretraining. How would VE perform when pretraining from scratch for 300/800/1600 epochs? Also, more datasets (e.g., ImageNet-21k) and more backbones should be included to show the scalability of the framework.

**Questions:**

Can you discuss more about how your work may influence a larger community than self-supervised visual representation learning, such as foundation models or multimodal learning?

---

### Official Review · Reviewer_jdSs · 2023-11-01

**Soundness:** 3 good
**Presentation:** 3 good
**Contribution:** 2 fair
**Rating:** 3
**Confidence:** 4

**Summary:**

This paper introduces ELECTRA from NLP to the Vision domain.  To enhance the diversity of generated patches, the authors inject random noise into the generator. To train the discriminator, the authors employ both patch-level and image-level discrimination. Different from previous work like MAE, the proposed method uses the discriminator rather than the generator as the pre-training outcome. Experimental results verify the proposed method achieves comparable performance with MAE, and SimMIM.

**Strengths:**

* The writing is clear and easy to follow.
* The injection of noise makes sense since there exists the one-to-many mapping for the generator.
* Many visualizations are provided to make a better understanding.

**Weaknesses:**

* The essential contribution is to introduce ELECTRA to the vision domain, which is quite limited. Although the authors propose the noise injection and the patch-level & image-level discriminator, however, 1) the noise injection brings marginal performance gain as demonstrated in Table 3, 2) the patch-level & image-level discriminator has been well-studied in image GAN, which is also missed in the related works.
* Experimental comparison is not convincing, the baselines (i.e., MAE and SimMIM) are behind the times (CVPR 2022).

**Questions:**

* What are the advantages of introducing ELECTRA to the vision domain? Is there any unique superiority compared to the existing pre-training scheme like SimMIM?

---

### Meta-Review · Area_Chair_4Vtz · 2023-12-10

**Metareview:**

This paper adapts ELECTRA (an NLP pretraining strategy) to the vision domain.  ELECTRA first masks parts of input texts and trains a generator and discriminator to reconstruct the texts and identify which parts are original or replaced.  This paper adapts with additional tricks such as random latent noise injection and a hierarchical discrimination loss.

The strengths of the paper are clear writing and a reasonable adaptation of an NLP strategy to vision.  The weaknesses of the paper are unconvincing experimental validation and lack of technical novelty.

The paper has received 3 reviews and ratings of 5/3/5.  No rebuttals are provided.

The AC recommends rejection.

**Justification For Why Not Higher Score:**

All the reviewers question the experimental validation and performance gains.  There are no rebuttals.

**Justification For Why Not Lower Score:**

N/A

---

### Decision · Program_Chairs · 2024-01-16

Reject